# Investigating the Effect of Urbanization on Weather Using the Weather Research and Forecasting (WRF) Model: A Case of Metro Manila, Philippines

**Jervie M. Oliveros [1,2,\*], Edgar A. Vallar [2] and Maria Cecilia D. Galvez [2]**

1. Physics Department, Technological University of the Philippines, Manila 1000, Philippines
2. Applied Research for Community, Health, and Environment Resilience and Sustainability (ARCHERS), Center for Natural Science and Environment Research (CENSER), Physics Department, De La Salle University, 2401 Taft Ave., Malate, Manila 1004, Philippines; edgar.vallar@dlsu.edu.ph (E.A.V.); maria.cecilia.galvez@dlsu.edu.ph (M.C.D.G.)
\* Correspondence: jervie_oliveros@tup.edu.ph; Tel.: +63-02-372-8933

**Abstract:** The effect of urbanization of Metro Manila, particularly on the amount of sensible heat flux, rainfall and temperature of selected urban and rural areas, was investigated using the Weather Research and Forecasting Version 3.4.1 (WRFV3.4.1) model. National Center for Environmental Prediction - Final (NCEP-FNL) grib1 data from 2000 to 2010 were used as inputs into the model for meteorological data. The Mann–Kendall trend test (M–K test) was utilized to verify the significance of the trends while Sen's slope estimator was used to quantify the measured trends. Results showed that, on average, the sensible heat flux of Metro Manila is about $1.5 \times 10^8$ Jm$^{-2}$ higher than in selected areas outside Metro Manila. The occurrence of an urban heat island (UHI) effect was detected in Metro Manila by comparing the difference in the minimum and maximum temperatures. For the selected urban and rural areas, the minimum and maximum temperature differences (relative to Metro Manila) are around 0.4 to 2.4 °C and 0.83 to 2.3 °C, respectively. Metro Manila recorded higher 11-year average values of rainfall during the summer season (8% to 64%), rainy season (15% to 305%), and transition season (8% to 232%) when compared with selected areas from 25 to 100 km from Manila. These results show that the sensible heat flux, temperature and rainfall in Metro Manila is affected by Metro Manila's urbanization.

**Keywords:** WRF; urbanization; sensible heat flux; rainfall; temperature; Mann-Kendall Test

## 1. Introduction

Urbanization is a direct result of physical growth of urban areas due to an increase of their populations. Based on the 2018 Revision of World Urbanization Prospects, the global urban population is expected to exceed the rural population by year 2050. By that time, the urban population will be approximately 70% of the global population and more than half of will be concentrated in Asia [1]. The higher demands of housing in urban areas bring changes in land use where vegetation areas are transformed to high rise buildings and roads. Sensible heat flux refers to the amount of energy transferred to the environment by the modified urban surface and the incoming solar radiation. Sensible heat flux can alter the energy balance and, eventually, the temperature of cities. This can produce an urban heat island (UHI) effect in cities. UHI is the warming of an urban area which can be observed in the decrease of the diurnal temperature range (DTR). DTR is the variation between the maximum temperature (daytime) to the minimum temperature (nighttime) [2]. All of these phenomena are causing noticeable changes in meteorological conditions, and climates of cities and nearby areas [3].

The energy balance of cities is accidentally altered by human activities [4,5]. There is an increase in energy consumption due to the numbers of vehicles, and aerosol gases in the atmosphere of urban areas has the effect of decreasing surface insulation [6]. An increasing difference between the amount of absorbed radiation to the reflected radiation of asphalted road surfaces and concrete high-rise building walls may cause turbulent convection [5].

The common materials that make up an urban area are concrete, asphalt, and brick, which are very different from natural materials of rural areas, such as trees, grass, and wet soils. Heat capacities and conductivities of asphalted roads and concrete walls of buildings are large compared to wet soil and grassland. The ability of urban materials to absorb more solar radiation makes urban areas warmer than rural areas. The noticeable decrease in the observed temperature difference of cities during daytime and nighttime, in addition to the difference between the air and surface temperature of central areas of the city and rural areas, are some of the effects of urbanization [2]. This anomaly in temperature relates to the urban heat island (UHI).

Urbanization may increase sensible heat flux, reduce evapotranspiration and create a deeper atmospheric boundary layer [7–9]. These altered conditions of the atmosphere may bring less water vapor, a high water-vapor mixing ratio, and reduce convective available energy for initiation of rainfall [10].

Metro Manila or the National Capital Region (NCR), the capital of the Philippines, is located at 14.59° N, 120.98° E. It is considered one of the top five largest urban areas in the world, having an estimated population of 13,000,000 as of 2015, and projected growth of 1.5% annually [11]. It has a land area of 638 km$^2$ (around 0.20% of total land area of the Philippines) and a density of about 21,000 people per km$^2$. Due to its limited size, coastlines of Manila Bay, Laguna Bay, and Pasig River, and green parks were converted to high-rise building (commercial/residential), with asphalted roads. Large portions of the coastline are shelters for marginalized urban dwellers. In this study, NCR has been classified as an urban area (2000–2010). Most of the places in Metro Manila are below sea level, so most of its population are affected by floods brought by typhoons, sea level rise and local thunderstorms. In addition, poor ventilation, particularly in depressed areas, will bring discomfort and can cause health-related problems. Therefore, studying the effect of urbanization of Metro Manila on its weather, particularly on sensible heat flux, temperature and rainfall, using regional climate modeling, is essential.

## 2. Materials and Methods

### 2.1. Data

The study utilized meteorological data from January 2000 to December 2010 of the National Center for Environmental Prediction-Final (NCEP-FNL) grib1 global analysis data. The grid resolution of the data is 1° × 1°, and it is prepared every 6 hours. The data consists of topographical and meteorological variables. The study period was chosen because major infrastructure was constructed within Metro Manila in this period. More infrastructure is currently being built so the current study is very significant.

The Weather Research and Forecasting Model or WRF is a mesoscale numerical weather prediction system intended for both research and operational applications. The development of this model is an effort by multiple agencies aiming to build a forecast model and data assimilation system, to enhance understanding and advance weather prediction [12]. Wang et al. (2006) [13] stated that WRF performs better than the Fifth-Generation Pennsylvania State University (PSU) / National Center for Atmospheric Prediction (NCAR) Mesoscale Model or MM5 model in simulating the position and movement of weather systems that lead to the formation of precipitation processes. A comparative study of WRF and Regional Climate Model (RegCM) to predict weather in the upper Mahaweli River basin was conducted by Mafas et al. (2016) [14]. The result showed that rainfall value was more accurately predicted in the area by the WRF model than the RegCM model. A similar

result was obtained by Gsella et al. (2014) [15], when they evaluated the performance of MM5, WRF and Tropospheric Realtime Applied Meteorological Procedures for Environmental Research (TRAMPER) models.

The Weather Research Forecasting Preprocessing System (WPS) of the WRF consists of 3 executable files namely: geogrid, ungrib, and metgrid. The main function of geogrid is to define the topographic features of the selected domain. The ungrib reads and rewrites the Network Common Data Form (NetCDF) or grib file formats into intermediate files for the model to easily understand. Then, the metgrid horizontally interpolates the meteorological parameters in the domain. The output of metgrid will be used by the WRF as input for generating real or ideal cases.

The Weather Research and Forecasting Version 3.4.1 (WRFV 3.4.1) version of the model was used in the study with the physics options enumerated in Table 1. Listed in this table are the compatible physics schemes suited to the domain after the sensitivity analysis were made. One-way nesting with 27 km × 27 km and 9 km × 9 km grid spacings (the availability of the resources is the reason of choosing this resolution) is used in simulating the Luzon as parent domain and Metro Manila as nested domain, respectively. To reduce the simulation time, the model was set to write the output every 3 hours. The accumulated sensible heat flux, temperature at 2 m above the surface and rainfall were extracted in the output. Validation of the model output (Temperature and Rainfall) were conducted using the historical data of the Philippine Atmospheric Geophysical and Astronomical Services Administration (PAGASA), the weather bureau of the Philippines. It showed that, using the schemes listed in Table 1, both generated temperature and rainfall of the selected areas were close (5–15% differences) to the observed data. The differences may be due to the resolution and data assimilation technique of the meteorological (NCEP-FNL Reanalysis) data used in the study.

**Table 1.** Physics schemes used in this study.

| Microphysics | Weather Research and Forecasting (WRF) Single Moment 3 Class Scheme |
|---|---|
| Shortwave and Longwave Radiation | Dudhia Scheme and Rapid Radiative Transport Model (RRTM) Scheme |
| Planetary Boundary Layer | Yonsei University (YSU) Scheme |
| Cumulus Physics | Kain-Fritsch Scheme |
| Land Surface Physics | Noah Land Surface Model (LSM) Scheme |
| Initial and Boundary Condition | National Center for Environmental Prediction – Final (NCEP-FNL) Reanalysis |

The Philippines' climate can be divided into a dry season and a wet season. The weather of December, January and February (DJF) is considered the cool and dry season. This is because the wind originates from the high latitude regions and moves towards the northwest Pacific Ocean and Philippines. This air stream is also known as the Northeast Monsoon or "Amihan". March, April and May (MAM) is a transition period where hot and dry winds come from high pressure systems in the Pacific Ocean. These months coincide with the local summer season. The months of June, July and August (JJA) consist of hot and moist air which brings more rainfall. The prevailing wind during this period is the Southwest Monsoon or "Habagat". Another transition period starts in September and ends in November (SON) but, in this case, hot and wet winds come from the West Philippine Sea. In addition, tropical cyclones contribute largely to the rainfall in the Philippines and occur mostly during JJA and SON.

The Modified Coronas Classification (MCC) was used to obtain the weather classification of each selected site. A Type I climate is characterized by a defined dry and a wet season; wet from June to November and dry, the rest of the year. For Type II, there is no dry season. On the other hand, a Type III climate is described by not very pronounced seasons, dry from November to April and wet for the rest of the year. This resembles Type I climate since they both have a short dry season. Most of the sites selected in this work are Type I. Type II sites in this study are Infanta (75 km south), Lucena (100 km

South) and Dingalan (100 km east). These 3 sites have been separated from the east and south plots since their climate is very different from that of Type I and III. Type III sites studied were Cabanatuan (100 km north) and Luisiana, Laguna (75 km south) but these were included in the south and north plots since their climate is like that of Type I.

The temperature from 2000–2010 was divided into 3 parts, namely: the minimum temperature ($T_{min}$), the maximum temperature ($T_{max}$) and the diurnal temperature range (DTR). $T_{max}$ used in the study is the temperature generated by the model every 06:00 UTC or 02:00 PST (Philippine Standard Time). $T_{min}$ is the temperature processed by the model every 21:00 UTC or 05:00 PST. The diurnal temperature range or DTR is the difference between $T_{max}$ and $T_{min}$. To identify the presence of the urban heat island effect (UHI), the DTR of urban area and rural areas was observed.

To recognize trends, the Mann–Kendall (M–K) test was conducted. This test is a statistical method used to identify trends in data, detect climate change, and, in particular, detect UHI intensity [16,17]. Its procedure is based on ranking and is therefore resistant to results of extreme values and deviation from a linear relationship [18]. This generally measures the correlation between two variables and requires the data to be independent, and can accept outliers in the data. The standardized test statistics Z will indicate whether there is an increasing or decreasing trend in the time series. To quantify the change in magnitude of the trend per unit time, Sen's slope estimator was combined in the test [16]. The World Meteorological Organization (WMO) highly recommended the use of these tests (M–K test and Sen's test). Thus, many researchers have used either both the M–K test and Sens' test, or only the M–K test, to investigate the data's trend and its magnitude [19–23].

## 2.2. Domain

The three cities—namely, Manila, Pasig and Quezon City—were used to represent Manila data. These cities occupy approximately 40% of Manila's land area and almost 43% of its population lives in these areas. In addition, these cities were categorized by US Geological Survey (USGS) as urban and built-up areas during the period of study.

For urban–rural weather and climate comparison, the 19 nearby rural areas listed in Table 2 were selected in the study. The selected rural areas are in the big plain of Luzon Island (except for Tagaytay). Their topographies are similar to Metro Manila, so are ideal for weather comparison. The selection of the sites based on locations relative to Metro Manila were made to investigate if the leeward sites are most affected by the induced weather of Metro Manila. The coordinates, climate classification and land use category index of every location during the period of study are also included in the table. These were included to determine whether the urbanization of Metro Manila affects the local climate of nearby areas. Also, the range segregation (25, 50, 75, and 100 km) aids in determining the extent and magnitude of the effect of Metro Manila's urbanization.

Some of the surrounding rural areas in Bulacan, Rizal, Cavite and Laguna are also undergoing urbanizations. In this study, the urbanization of nearby areas is assumed to be negligible in comparison with the rate of urbanization of Manila. In addition, areas like Infanta, Dingalan and Lucena have been excluded in the plots in this subsection since they have a Type II climate (MCC). Tagaytay has also been excluded in the plots.

## 3. Results and Discussions

This section analyzes the urbanization of Metro Manila and its effect on the weather of Metro Manila and surrounding places (up to 100 km away), particularly in terms of sensible heat flux, temperature and rainfall. Graphs of these parameters were made to illustrate the changes over the period 2000–2010. Furthermore, Mann–Kendall results were conducted to show trends and their magnitudes.

## 3.1. Effects on Sensible Heat Flux

The materials (concrete, asphalt, and steel) that make up the numerous structures in Metro Manila absorb more solar radiation compared to natural materials (i.e., trees, wood, and soil) used

in neighboring rural areas. Part of the absorbed radiation is transformed into heat which causes an increase in the air temperature of cities [5,24]. Recent research on worldwide land use land cover (LULC) changes of urban areas suggests this may have contributed to increased global mean temperature. Hence, further studies on the impacts of LULC change on climate are necessary [25,26].

Table 2 lists the land use (LU) category assigned by the US Geological Survey (USGS) to Metro Manila and the other sites used in this study. Identifying the LU is important in assessing sensible heat flux, since the LU classification gives an idea of the predominant material in each site. Varied materials have different thermodynamic properties (such as heat capacity and emissivity).

**Table 2.** Selected urban and rural areas with corresponding Modified Corona Classification (MCC) and land use (LU) categories.

| Sites | Location | Coordinates | MCC Type | LU |
|-------|----------|-------------|----------|-----|
| | Quezon City | 14.68° N 121.02° E | I | 1 |
| Metro Manila (MM) | Manila | 14.60° N 121.02° E | I | 1 |
| | Pasig | 14.60° N 121.11° E | I | 1 |
| 25 km N | Sta. Maria | 14.82° N 120.95° E | I | 3 |
| | Bulacan | 14.79° N 120.88° E | I | 3 |
| 25 km E | Cogeo | 14.65° N 121.21° E | I | 3 |
| 25 km S | Gen. Trias | 14.40° N 120.88° E | I | 3 |
| 50 km N | Angeles | 15.17° N 120.62° E | 1 | 3 |
| 50 km W | Mariveles | 14.45° N 120.55° E | I | 3 |
| 50 km S | Gen. Aguinaldo | 14.19° N 120.80° E | I | 6 |
| | Calamba | 14.17° N 121.13° E | I | 3 |
| 75 km N | Gapan | 15.22° N 120.95° E | I | 3 |
| 75 km W | Olongapo | 14.83° N 120.33° E | I | 3 |
| 75 km S | Lipa | 13.94° N 121.16° E | I | 2 |
| | Luisiana (L) | 14.17° N 121.52° E | III | 6 |
| 100 km N | Cabanatuan | 15.50° N 120.97° E | III | 3 |
| 100 km S | San Pablo | 14.03° N 121.35° E | I | 6 |
| | San Juan | 13.76° N 121.41° E | I | 6 |
| | Infanta | 14.73° N 121.65° E | II | 6 |
| Elevated Area | Tagaytay | 14.15° N 120.98° E | I | 6 |
| | Lucena | 13.94° N 121.61° E | II | 6 |
| | Dingalan | 15.42° N 121.32° E | II | 6 |

LU 1: Urban and Built-up, 2: Dryland Cropland, 3: Irrigated Cropland and 6: Cropland/Woodland Mosaic. (Source: US Geological Survey (USGS) 24 Land Use Classification).

Figure 1 shows the total sensible heat flux of January 2005 in Luzon. During this period, no rainfall was recorded in all the study sites. The sensible heat flux of Metro Manila is represented by the red region while the yellow regions are other highly populated places, some of which are located close to the mountains found in Luzon and neighboring rural areas of Metro Manila. The terrain of Luzon is also illustrated in the figure. This reflects the fact that some factors affecting sensible heat flux of an area are topography (areas near the mountain) and land use (Manila).

In determining the detailed effect of urbanization on the amount of sensible heat flux, the annual average sensible heat flux of Metro Manila and selected places (up to 100 km away from Manila) was measured and analyzed. A comparison of the sensible heat flux of Metro Manila with selected places south and north of Manila are plotted in Figures 2a–e and 3a–e, respectively. These show that, on average, sensible heat fluxes of selected rural areas from north and south obtained a value closer to $1.0 \times 10^8$ Jm$^{-2}$. However, the average sensible heat flux of Metro Manila is about $2.5 \times 10^8$ Jm$^{-2}$; that is, Metro Manila has a higher sensible heat flux compared with selected rural areas located north and south of Metro Manila. This result shows that the urbanization of Metro Manila causes it to have a higher sensible heat flux since the amount of incoming solar radiation is not significantly different for all locations considered in this study.

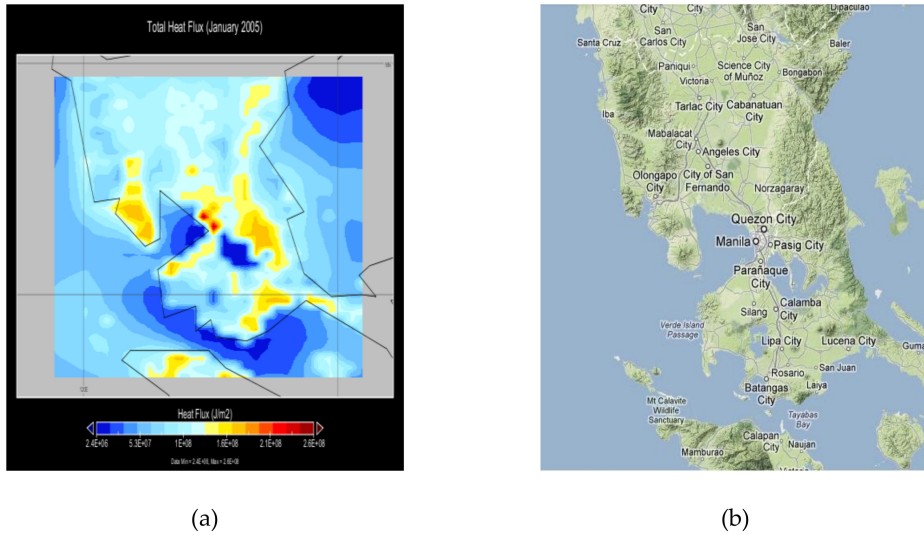

**Figure 1.** (**a**) Variation of sensible heat flux (in Jm$^{-2}$) and (**b**) terrain of Luzon from Google maps.

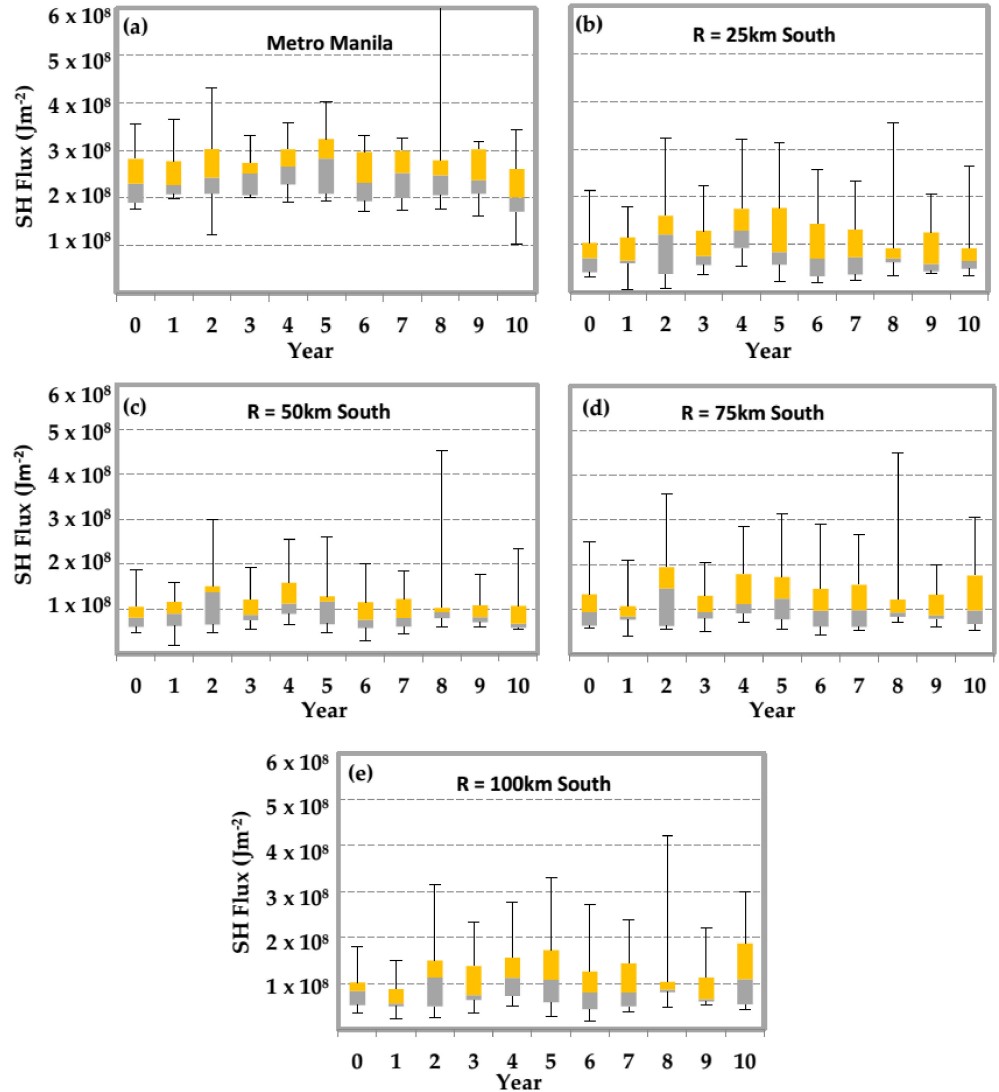

**Figure 2.** Annual variation of sensible heat (SH) flux of Metro Manila (**a**) and selected places in the south of Manila (**b**–**e**) for the period 2000–2010. Figure 2a is the same as Figure 3a.

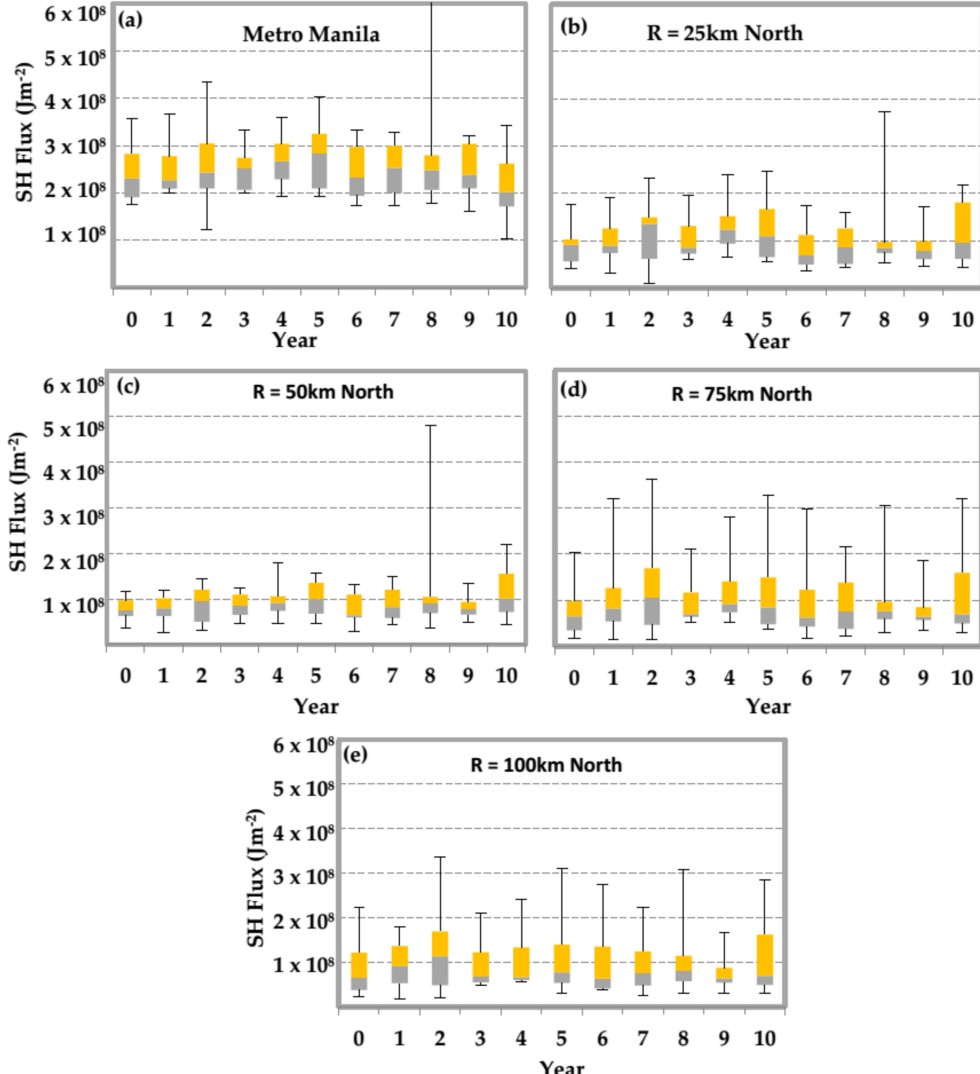

**Figure 3.** Annual variation of sensible heat (SH) flux of Metro Manila (**a**) and selected places in the north of Manila (**b–e**) for the period 2000–2010. Figure 3a is the same as Figure 2a.

It can be seen in Figures 2a–e and 3a–e that there is no significant difference in the magnitude of the average sensible heat flux when comparing sites north and south of Manila. Similar results were obtained for areas east and west of Manila. Further, even the sensible heat flux of selected areas 25 km from Manila was almost the same as for all directions and all distances away from Manila. The 25 km area seems unaffected by the urbanization happening in Metro Manila. Therefore, the urbanization of Metro Manila only affects its sensible heat flux. Moreover, for the period of the study (2000–2010), Manila's sensible heat flux has remained constant at about $2.5 \times 10^8$ Jm$^{-2}$. This indicates that the level of urbanization of Metro Manila, as measured using sensible heat flux, has leveled off for the 2000–2010 period. There may be more development taking place in the area, but the sensible heat flux parameter does not indicate this detail.

### 3.2. Effects on Rainfall

Metro Manila possesses repetitive structures such as buildings and its roads are constructed using concrete or a combination of asphalt and concrete. The high heat capacities of these materials contribute to sensible heat flux and then increase the temperature of the area. In addition, buildings can provide air convergence, pushing warm and moist surface air into the cooler air above it. This can modify local atmospheric conditions that lead to less water vapor, more water-vapor mixing and less

available energy for generating convection and rainfall formation [27]. Therefore, investigating the impact of urbanization on rainfall is also important.

### 3.2.1. Annual Variation of Monthly Rainfall

Figure 4 shows the time series of the monthly accumulated rainfall of selected urban and rural areas north and south of Manila for the period 2000–2010. This figure illustrates the durations of dry and wet seasons, with large volumes of rainfall measured for the month of June up to November, and rainfall falling below 500 mm for the remaining months. Similar characteristics were obtained for the selected sites categorized by MCC as Type I or Type III.

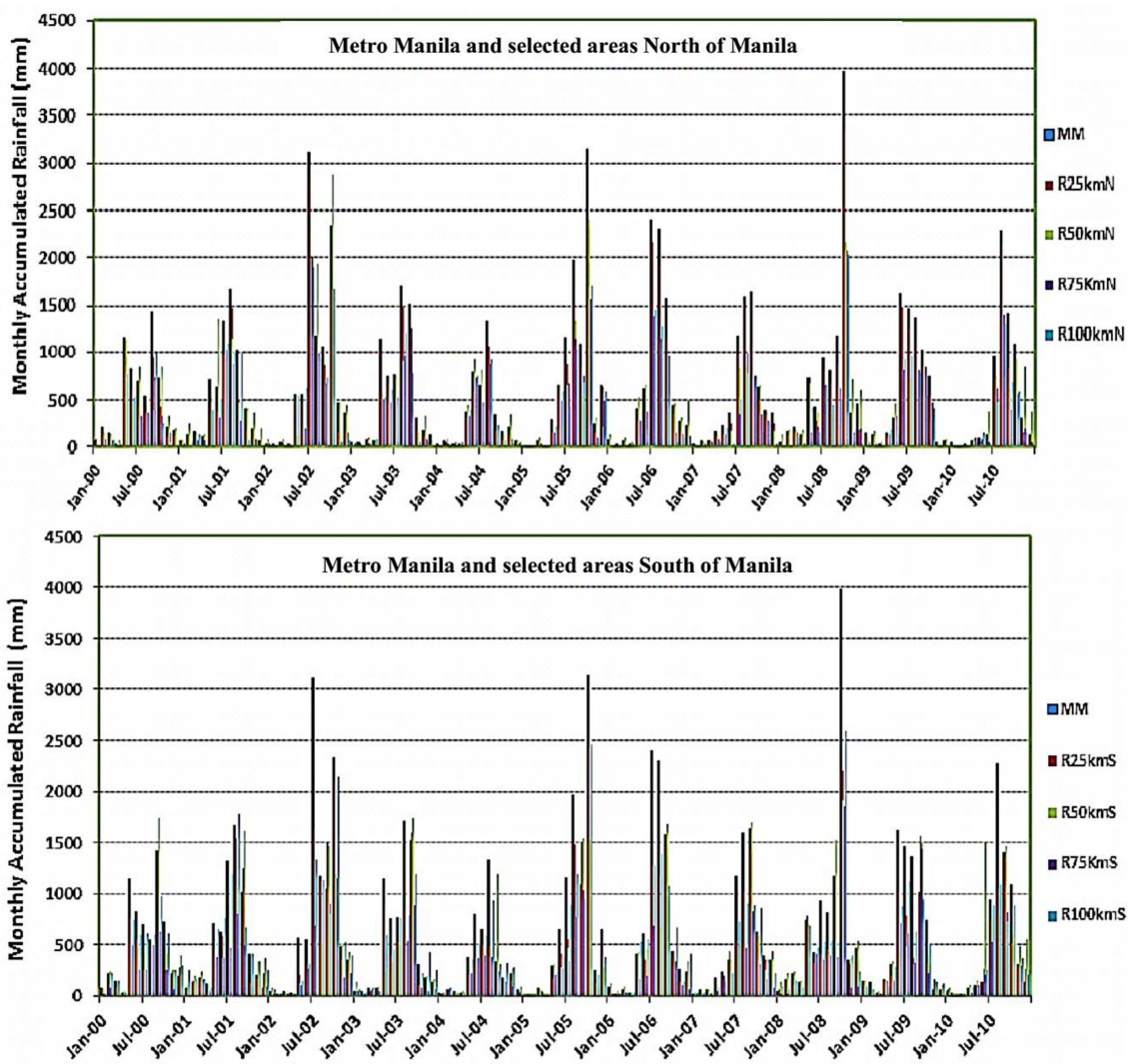

**Figure 4.** Rainfall distribution (in mm) of Metro Manila and selected areas north and south of Manila for 2000–2010.

A different result was observed for rainfall distribution for Tagaytay, Infanta, Lucena and Dingalan. Tagaytay is situated hundreds of meters in altitude above sea level, and the air temperature and air density are affected by the low pressure. The condensation rate is faster at high altitudes than at lower altitudes. Therefore, rains are more frequent, even during the dry season of the year. Further, there was no dry season in Infanta, Lucena and Dingalan for most of the 11-year period. This is because these places have a Type II climate classification.

3.2.2. Average Seasonal Rainfall

The variations of average seasonal rainfall per year for the selected sites are plotted in Figure 5. This figure provides information on how urbanization may affect rainfall during either the dry seasons (DJF and MAM) or the wet seasons (JJA and SON). Looking at Figure 5a, the 50 km groups (south, north, and west) and 100 km south recorded average rainfalls of 158.77 mm, 136.10 mm, 237.09 mm and 126.10 mm, respectively. These are about 5% to 98% larger than the rainfall of Metro Manila. However, the rainfall of Metro Manila is approximately 4% to 220% greater than the 25 km group (south, north, and east), 75 km group (south, north, and west) and 100 km north.

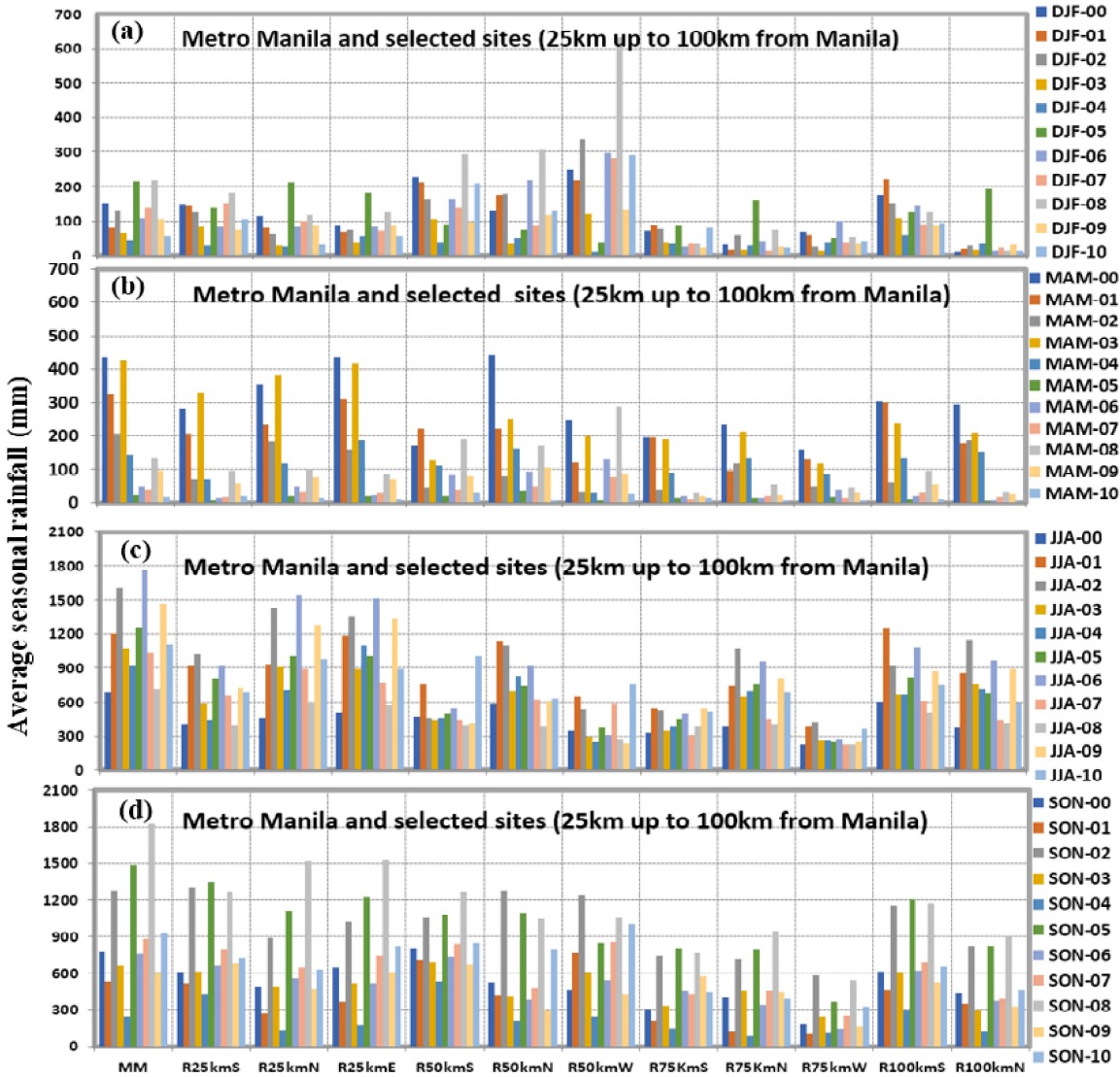

**Figure 5.** Volume of rainfall (mm) for 11 years (2000–2010) during the dry seasons (December, January, and February (DJF) and March, April, and May (MAM)) and wet seasons (June, July, and August (JJA) and September, October, and November (SON)) of the selected sites.

In addition, Figure 5a also suggests that the average seasonal rainfall during the Northeast Monsoon is bigger for sites south (25 km, 50 km, 75 km and 100 km) of Manila compared with sites north of Manila (25 km, 50 km, 75 km and 100 km). This result agrees with the findings of Huff and Vogel (1978) [28] during their Metropolitan Meteorological Experiment (METROMEX) project conducted in St. Louis, Missouri, and Estoque and Sta. Maria (2000) [27] in Metro Manila. It can be

concluded that the effect of urbanization on rainfall is more visible downwind of the urban area in relation to the prevailing wind.

A decreasing trend can be observed for the rainfall during the summer season (MAM) of 2000–2010 for all selected places (Figure 5b). The 11-year average rainfall of Metro Manila is 172.46 mm for this season and is the highest compared with all the sites having similar climate classification. This corresponds to 8% to 38%, 15% to 41%, 51% to 64% and 33% to 41% higher than the rainfall values in the groups of 25 km, 50 km, 75 km and 100 km, respectively.

This observed high value of rain in Metro Manila during summer might be the urbanization effect on rainfall in urban areas. Studies have shown that modifying geometry of urban areas can generate its own climatic features resulting in an increase in the number of urban-induced changes in rainfall [10,29]. Subbiah et al. (1991) [5] stated that urban areas experienced more rainy days and more thunderstorms during warm seasons. Furthermore, rainfall patterns of the urban area are modified due to the combination of atmospheric circulation induced by UHI and aerodynamic effect of the urban canopy [30].

Historically, Type I and III (MCC) places experienced more rain during the rainy season (JJA) and the transition period (SON). As suggested in Figure 5c, Metro Manila, along with the 25 km, 50 km, and 75 km north sites, received higher 11-year average rainfall during Southwest Monsoon (JJA) season compared with the transition (SON) season. Further, during JJA, the rainfall values of the 25 km, 50 km and 75 km north sites are greater than the south sites. However, during SON (see Figure 5d), the south sites have greater rainfall than the north sites. Moreover, the rainfall for the south sites is greater during SON compared to the JJA period. This is because the rain brought by the Southwest Monsoon or Habagat (JJA) is intensified by tropical cyclones (mostly coming from the Pacific Ocean) which frequently make landfall in the North Luzon. For the South Luzon, a considerable number of tropical cyclones arrive during the transition period or SON.

Among all the sites selected in this work (except 100 km north), Metro Manila experienced a higher 11-year average rainfall during JJA (1166 mm) and SON (906.75 mm). These are about 15% to 305% and 8% to 232% higher during JJA and SON, respectively. In Figure 5c, it is noticeable that the magnitudes of rainfall decrease as the distance from Manila increases, except for the 100 km sites. The increasing number of aerosol gases due to human activities that serve an important role in cloud formation maybe the reason for such an increase in the magnitude of rainfall in urban areas. This so-called secondary effect is not considered in the study. The trends of precipitation are analyzed further in Section 3.3.4.

The value of rain during SON decreases from 25 km north up to 75 km north. No specific rainfall pattern can be deduced from the south sites.

In general, during summer (MAM) and rainy (JJA) seasons, the high values of average rainfall are found for areas located 25 km and 50 km north compared to 75 km and 100 km north. A similar observation can be made for SON and DJF: more rains occur in the areas under 25 km and 50 km south of Manila than in areas under 75 km and 100 km south. These observations may be the effect of urbanization on rainfall amount in nearby areas (25–50 km). In addition to this claim is the interaction between UHI occurrences in Metro Manila and sea breezes from Manila Bay and Laguna Bay.

*3.3. Effects on Temperature*

3.3.1. Maximum Temperature or $T_{max}$

The variation of the maximum temperature for the sites selected in this study for the period 2000–2010 is plotted in Figure 6. This graph shows that the maximum temperature is in the range of 23 °C to 35 °C. Excluding sites that have a different climate type (Infanta, Lucena and Dingalan) and elevation (Tagaytay), $T_{max}$ varies from 25 °C to 35 °C.

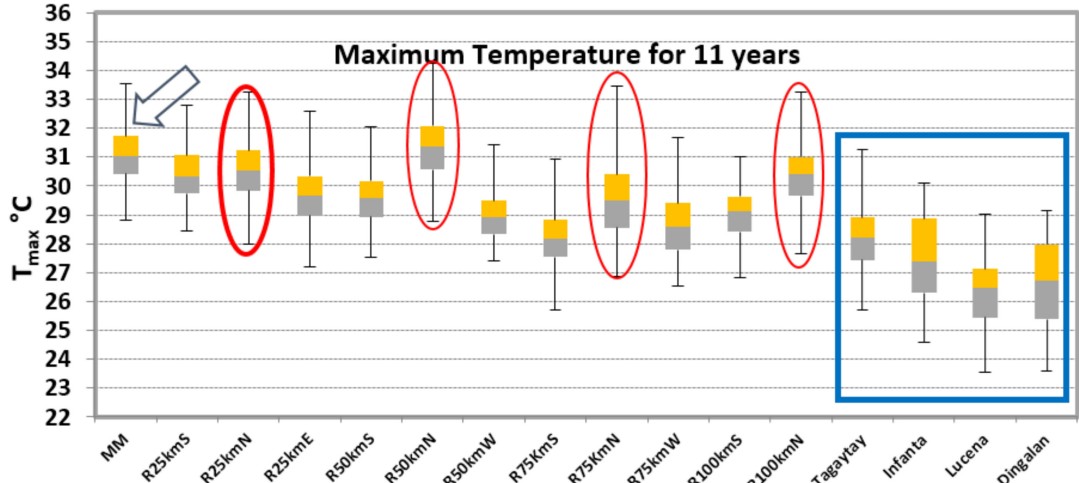

**Figure 6.** Spatial variation of temperature for the period 2000–2010. MM stands for Metro Manila (indicated by the arrow) while encircled are the north sites, and the blue box represents the elevated and MCC Type II.

As can be observed from Figure 6, the average $T_{max}$ of Metro Manila (about 31 °C) is larger than the $T_{max}$ of all the other sites selected except for Angeles City (about 31.5 °C) which represents the 50 km north dataset. The $T_{max}$ of Metro Manila corresponds to the average maximum temperatures of Manila, QC and Pasig. In addition, Angeles City can be considered a highly urbanized city due to conversion of an airbase into commercial, industrial and residential buildings. This is convincing evidence that pavements and vertical structures are linked to surface temperature increases during daytime. This brings discomfort and may cause serious health problems for urban dwellers [31].

Analyzing the values of the average $T_{max}$ by geographic location, the graph shows that the mean $T_{max}$ of north sites are higher compared to their counterpart equidistant places (south, west and east). The estimated difference between the averages (for the 11-year period) of $T_{max}$ of Metro Manila and the north sites (except 50 km N), south sites, east (Cogeo) and west are 0.83 °C, 1.73 °C, 1.5 °C and 2.3 °C, respectively.

### 3.3.2. Minimum temperature or $T_{min}$

The minimum temperatures, $T_{min}$, of selected urban and rural areas for the period 2000–2010 are shown in Figure 7. Just like the results of $T_{max}$, the highest average $T_{min}$ is recorded in Metro Manila compared with all the other sites selected. The sites selected have an average $T_{min}$ that is lower than the average $T_{min}$ of Metro Manila, in the ranges of 0.4 °C to 2.4 °C. The sites included in the blue box have the same variation in $T_{min}$ as the other sites and their average $T_{min}$ is mostly lower than the average $T_{min}$ of the other sites.

Sea breezes from Manila Bay (west of Manila) and Laguna Bay (south of Manila) greatly affect the temperature of Metro Manila. Sea breeze occurs due to the temperature gradient between a body of water and the land surface. This can bring a cooling effect to places near the coastline area [32]. However, if this wind circulation has a cooling effect for areas near this body of water (Metro Manila and south of Manila), results show the opposite. Even if bodies of water are close to Metro Manila and sites up to 25 km south of Metro Manila, the $T_{min}$ of these sites are greater than the sites located inland.

All the factors that might alter the weather, such as wind systems, periodic warming and cooling pattern in the tropical Pacific Region or El Niño Southern Oscillation (ENSO), and many more, are included in the results. Thus, this suggests that urbanization, which is occurring in Metro Manila, affects its minimum temperature.

Comparing the different sites outside of Metro Manila, Figure 7 also shows that places south of Manila recorded the highest average minimum temperature compared with north sites except 75km

south (dotted red circle). Among the south sites, 25 km south reached the highest average value of $T_{min}$ (24.3 °C). Moreover, Mariveles (50 km west) recorded a 24.8 °C average $T_{min}$, which is only 0.4 °C lower than the $T_{min}$ of Metro Manila.

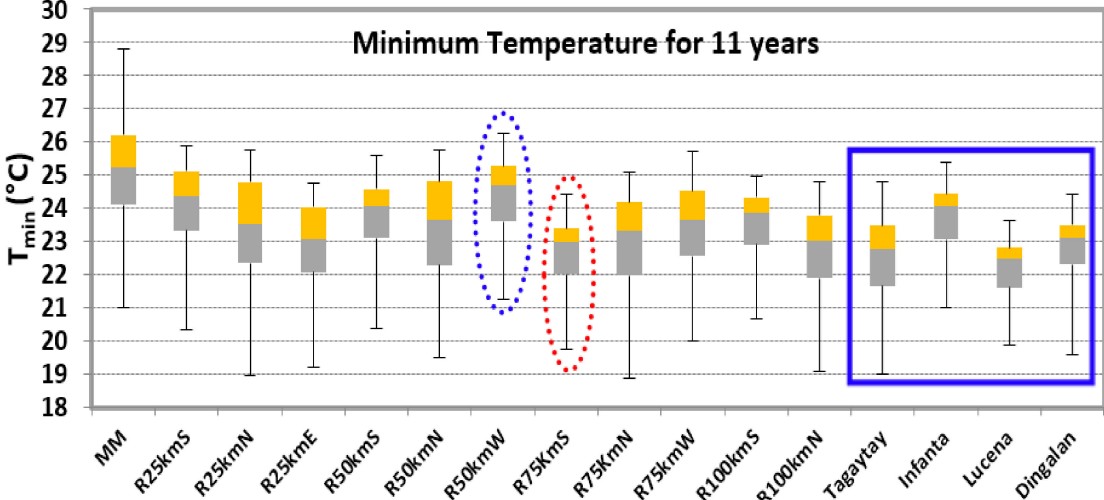

**Figure 7.** Spatial variation of $T_{min}$ (°C) for the period 2000–2010. The blue dotted line represents Mariveles while the blue box separates the sites that are elevated and MCC Type II.

### 3.3.3. Diurnal Temperature Range (DTR)

Results of $T_{max}$ and $T_{min}$ successfully revealed the occurrence of UHI and the effect of urbanization on temperature. The diurnal temperature range analysis is incorporated to give additional proof in illustrating the UHI effect.

Figure 8 shows the monthly average DTR for the period 2000–2010. Most of the high and low peaks of DTR happened during the summer season and Southwest Monsoon, respectively. Only 25 km north and south are included in the graph of DTR trend of Metro Manila because of the suspicion that the urbanization effect of Metro Manila only extends up to those areas.

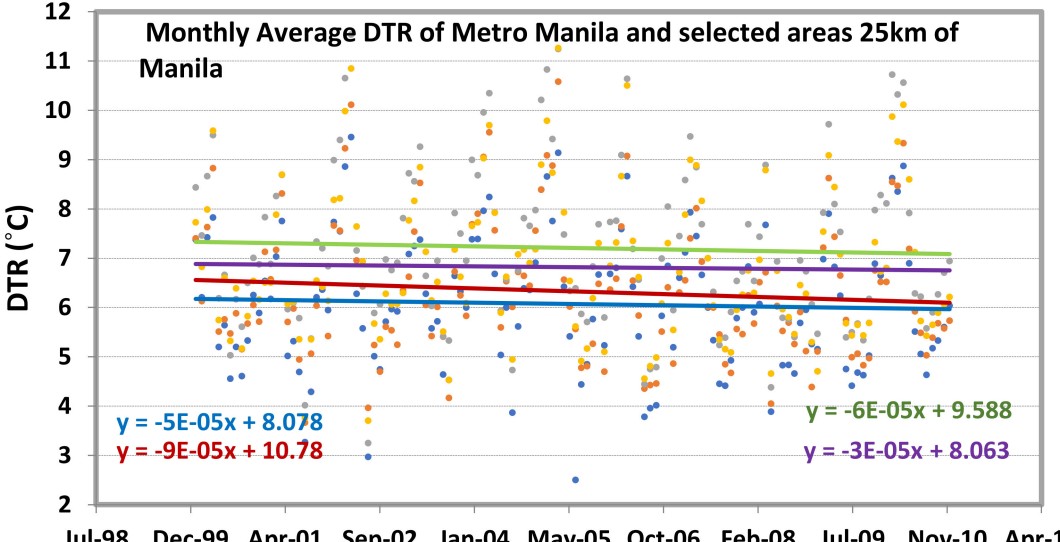

**Figure 8.** Decreasing diurnal temperature range (DTR, °C) of Metro Manila (blue) and 25 km south (red), 25 km north (green) and 25 km east (violet) of Manila. Decreasing rate is largest at 25 km south.

Decreasing trends were measured for Metro Manila, 25 km north, 25 km south and 25 km east of Manila. This is a proof of how Metro Manila affects the temperature of nearby areas. Due to the

closeness of the magnitudes of the slopes of 25 km sites, it is not possible to identify the direction in which the effect of urbanization of Metro Manila is at its greatest. A decreasing DTR means that either $T_{min}$ is increasing (at a faster rate compared to $T_{max}$) or $T_{max}$ is decreasing (approaching $T_{min}$). Both cases mean that there is not much difference between $T_{max}$ and $T_{min}$ values, that is, the temperature at 05:00 PST ($T_{min}$) is close to the temperature at 14:00 PST ($T_{max}$). The decreasing rate of DTR of Metro Manila is more evidence (aside from $T_{min}$ (urban) > $T_{min}$ (selected sites)) of the occurrence of the urban heat island (UHI) effect.

In general, all the selected sites have shown decreasing trends of DTR. However, it seems that no trend exists since the line is almost horizontal (approaching zero slope) or the DTR is constant over time (2000–2010).

### 3.3.4. Trend analysis using M–K test and Sen's Slope

The Sen's slope analysis results for $T_{max}$ and $T_{min}$, DTR, and rainfall computed for the period 2000–2010 are listed in Table 3. It shows that there is increasing trend (slope is positive) of $T_{min}$ and $T_{max}$ for all selected places except for the $T_{max}$ in 75 km west of Manila. The values of the slopes of $T_{max}$ range from $-0.001$ °C/year to $0.002$ °C/year, while for $T_{min}$ are from $0.001$ °C/year to $0.003$ °C/year.

On the other hand, there is a decreasing (negative slope) trend for DTR. This implies that the minimum temperature increases at a faster rate than the maximum temperature. Thus, the difference between $T_{max}$ and $T_{min}$ decreases yielding a negative slope/trend. This is the same as the result of Karl, T. R. et al. (1991) [33], and means that all the selected locations are experiencing urban growth and land use change. The areas 25 km south, 50 km south (Calamba and Gen. Aguinaldo) and 75 km west of Manila have a larger rate (more negative) while other places have almost the same rate as Metro Manila. Lower rates are found for 25 km east (represented by Cogeo) and 100 km south (represented by San Pablo, Laguna and San Juan, Batangas).

Furthermore, the average slopes of rainfall for the selected areas under similar weather classification are also included in the same table. Trends in rainfall for all the selected sites are slightly positive (less than 1 mm/year) for the period 2000–2010. This means that there has been only a slight increase in rainfall during this period. The areas 50 km west and 50 km south of Manila have a higher slope compared with all the other sites, including Metro Manila. This may be because of the industrial park (50km south) and increasing population resulting in increased convection. However, upon verification of $T_{max}$ and $T_{min}$, DTR, and rainfall trend using the M–K test (Table 3), the computed p-values are larger than 0.05. Thus, it can be concluded that the computed trends are not statistically significant for the 11-year period under consideration in this study.

One probable reason for a statistically insignificant result is the period (2000–2010) of study. Perhaps longer datasets are necessary to see if the trends persist and to allow further investigation of urbanization effects on weather. In addition, Metro Manila was already categorized as LU = 1 in 2000, so the transition of Metro Manila from rural to urban was not captured in the study.

**Table 3.** Mann–Kendall results for the period 2000–2010.

| | | | | | | | | | | | | |
|---|---|---|---|---|---|---|---|---|---|---|---|---|
| **Mann Kendall Test and Sens Slope Estimate on 11 Years Rainfall** | | | | | | | | | | | | |
| Locations | MM | 25 km S | 25 km N | 25 km E | 50 km S | 50 km N | 50 km W | 75 km S | 75 km N | 75 km W | 100 km S | 100 km N |
| Kendall's Tau ($\tau$) | 0.013 | 0.024 | 0.023 | 0.02 | 0.046 | 0.006 | 0.073 | 0.019 | 0.033 | 0.019 | 0.019 | 0.029 |
| *p*-value (two-tailed) | 0.829 | 0.685 | 0.704 | 0.74 | 0.483 | 0.922 | 0.214 | 0.584 | 0.575 | 0.754 | 0.746 | 0.627 |
| Sen's Slope (Q) | 0.128 | 0.125 | 0.164 | 0.183 | 0.341 | 0.061 | 0.728 | 0.091 | 0.125 | 0.076 | 0.172 | 0.083 |
| **Mann Kendall Test and Sens Slope Estimate on 11 Years Maximum Temperature ($T_{max}$)** | | | | | | | | | | | | |
| Locations | MM | 25 km S | 25 km N | 25 km E | 50 km S | 50 km N | 50 km W | 75 km S | 75 km N | 75 km W | 100 km S | 100 km N |
| Kendall's Tau ($\tau$) | 0.056 | 0.021 | 0.035 | 0.049 | 0.01 | 0.046 | 0.038 | 0.019 | 0.02 | −0.031 | 0.054 | 0.046 |
| *p*-value (two-tailed) | 0.353 | 0.721 | 0.558 | 0.402 | 0.871 | 0.431 | 0.521 | 0.732 | 0.735 | 0.601 | 0.365 | 0.439 |
| Sen's Slope (Q) | 0.002 | 0.001 | 0.001 | 0.002 | 0 | 0.002 | 0.001 | 0.001 | 0.001 | −0.001 | 0.002 | 0.002 |
| **Mann Kendall Test and Sens Slope Estimate on 11 Years Minimum Temperature ($T_{min}$)** | | | | | | | | | | | | |
| Locations | MM | 25 km S | 25 km N | 25 km E | 50 km S | 50 km N | 50 km W | 75 km S | 75 km N | 75 km W | 100 km S | 100 km N |
| Kendall's Tau ($\tau$) | 0.076 | 0.077 | 0.048 | 0.06 | 0.089 | 0.043 | 0.091 | 0.043 | 0.049 | 0.047 | 0.075 | 0.067 |
| *p*-value (two-tailed) | 0.202 | 0.197 | 0.421 | 0.312 | 0.184 | 0.461 | 0.120 | 0.469 | 0.410 | 0.427 | 0.214 | 0.256 |
| Sen's Slope (Q) | 0.003 | 0.003 | 0.003 | 0.003 | 0.003 | 0.002 | 0.003 | 0.001 | 0.002 | 0.002 | 0.003 | 0.003 |
| **Mann Kendall Test and Sens Slope Estimate on 11 Years Diurnal Temperature Range (DTR)** | | | | | | | | | | | | |
| Locations | MM | 25 km S | 25 km N | 25 km E | 50 km S | 50 km N | 50 km W | 75 km S | 75 km N | 75 km W | 100 km S | 100 km N |
| Kendall's Tau ($\tau$) | −0.033 | −0.068 | −0.038 | −0.026 | −0.062 | −0.032 | −0.04 | −0.018 | −0.027 | −0.037 | −0.015 | −0.04 |
| *p*-value (two-tailed) | 0.572 | 0.259 | 0.542 | 0.655 | 0.291 | 0.593 | 0.497 | 0.684 | 0.645 | 0.530 | 0.805 | 0.499 |
| Sen's Slope (Q) | −0.002 | −0.003 | −0.002 | −0.001 | −0.004 | −0.002 | −0.002 | −0.002 | −0.002 | −0.003 | −0.001 | −0.002 |

Significant *p*-value = 0.05, $-1 \leq \tau \leq 1$.

## 4. Conclusions

This work successfully utilized the Weather Research and Forecasting (WRF) model to investigate the effect of the urbanization of Metro Manila on the weather of Metro Manila and the surrounding areas. Using WRF, meteorological parameters (such as the sensible heat flux, temperature and rainfall) for Metro Manila and selected sites 25 km, 50 km, 75 km and 100 km away from Manila were determined for the period 2000–2010. In addition, WRF also provided the USGS land use classification for all the sites in this study.

The difference in the sensible heat flux between Metro Manila and the selected areas in this study can be explained by the physical characteristics (such as increased heat capacities and a decrease in surface albedo) of the structures in Metro Manila. The sensible heat flux results show that the urbanization of Metro Manila does not affect the sensible heat flux of the surrounding areas. The urbanization of Metro Manila only affects its sensible heat flux.

Compared to the selected areas, Metro Manila attained the highest average $T_{min}$ of 25.2 °C. At 05:00 PST, Metro Manila is warmer than all the selected areas in this study. This is due to the UHI intensity which opposes the breeze coming from Manila Bay and Laguna Bay. UHI intensity is affected by land modification and population increases. This, in turn, can alter the energy budget and weather of the urban cities.

Comparing rural areas, 25 km, 50 km and 100 km south obtained high morning temperatures ($T_{min}$) while from 25 km to 100 km north obtained high noontime temperatures ($T_{max}$). Thus, the warming of Metro Manila extends only up to areas within 25 km of Manila, which is supported by the result of the DTR trend.

The differences in the temperatures of Metro Manila from the selected sites range from 0.4 °C to 2.4 °C and 0.83 °C to 2.3 °C for $T_{min}$ and $T_{max}$, respectively. One reason for this is the number of commercial and residential buildings that are scattered inside Metro Manila. Buildings increase surface roughness, leading to reduced ventilation rates in urban areas. Evapotranspiration is considered one of the major cooling processes for both atmosphere and the land surface. Transforming vegetation into high rise buildings in Metro Manila results in less evapotranspiration, larger sensible heat flux and a warmed environment.

Thus, these $T_{max}$ differences between Metro Manila and the other selected sites, along with the warming morning temperatures, are the effects of urbanization and indicate the presence of an UHI effect.

It is observed that over the period of study (2000–2010), the DTR values of Metro Manila and the selected sites were not affected by urbanization of Metro Manila.

Metro Manila's urbanization leads to increased surface runoff and, thus, produces less evaporation and evapotranspiration, increases surface temperature and sensible heat fluxes, and can generate more rainfall. That is why in most seasons (except during the Northeast Monsoon), the 11-year average rainfall in Metro Manila is larger than for all the selected sites (except at 100 km north during the transition season (SON)).

M–K tests show a positive trend for rainfall in Metro Manila and for all selected areas. However, since the computed *p*-values are greater than 0.05, the trend is not statistically significant.

**Author Contributions:** The authors shared their ideas in the conceptualization of this research. J.M.O conducted the formal analysis and investigation. The data curation, visualization, and supervision were made by E.A.V. and M.C.D.G. Finally, writing, review and editing of the manuscript were prepared by J.M.O and E.A.V.

**Funding:** This research was funded by, Commission on Higher Education (CHED)–Center of Excellence (COE) and the De La Salle University (DLSU) Science Foundation.

**Acknowledgments:** The researchers would like to acknowledge the support of Department of Science and Technology Science Education Institute (DOST-SEI) and Technological University of the Philippines (TUP), Commission on Higher Education (CHED) and to the valuable comments and suggestions of Dr. Gerry Bagtasa.

**Conflicts of Interest:** The authors declare no conflict of interest.

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
