# Peer review of "Investigating the Effect of Urbanization on Weather Using the Weather Research and Forecasting (WRF) Model: A Case of Metro Manila, Philippines"

_environments, doi:10.3390/environments6020010_

Round 1

Reviewer 1 Report

Comments:

This study measures the effect of urbanization of Metro Manila particularly to the amount of heat flux, rainfall and temperature. The paper investigates selected urban and rural areas. The Weather Research and Forecasting Version 3.4.1 (WRFV3.4.1) model was utilized using the NCEP FNL grib 1 meteorological data from 2000 to 2010. The Mann-Kendall trend test (M-K test) verified the significance of the trends and the Sen’s slope estimator quantified the measured trends. The author(s) suggests that the urbanization of Metro Manila affects its weather specifically heat flux, temperature and rainfall.

Overall, the paper has a unique approach to assess the impact of urbanization. However, the paper is lack of clear explanation of the reason of selecting Metro Manila, model selection and the results.

My concerns and comments are listed below.

1-      The paper needs a chapter on Metro Manila and the impacts of weather trends on this location. This chapter also needs to support the selection of cities, data points and variables.

2-      The paper should include a literature review, thus, the readers can identify similar studies which uses or not uses WRF, M-K test, Sen’s slope to analyze the effect of urbanization in different cities or regions. The comparison of results with other similar studies would also increase the credibility of the analysis.

3-      Table 1 does not explain much about the model. It needs to be summarized in detail for the reader.

4-      Figures 2 and 3 was not explanatory for the comparison of heat flux. The figures needs to be explained in details to support the author(s) argument. I think figure 2a and 3a are the same, this would confuse the reader.

5-      Figures 4 and 5 is hard to follow. They need detailed explanation.

6-      The main issue for the model, not having a control point for the analysis. I suggest to have a control city or region for this study, since the heat flux, rainfall and temperature impact might be a result of some other regions. A control region would help identify the trends that might not be the effect of urbanization.

7-      The table 4 suggests that trends for the selected variables are not statistically significant. Therefore, the data points were random and the argument on the impact of urbanization is not statistically significant. If the data were collected every 6 hours from 2000 to 2010, the data must be enough to identify the trend. Unless, the paper uses the annual averages in the analysis, I am not sure why 30 years of data were suggested by the author(s). This part should be discussed better, otherwise, the conclusion suggests no impact.

Author Response

please fine the response attached.

Reviewer 2 Report

General comments:

Investigating the effect of urbanization on weather is very interesting and actual topic and can be of great interest for readers. Unfortunately, I do have some reserves concerning the research design of this study. The study, which is presented in this manuscript, considers extensive investigation of the local climate at different locations in Philippines, but cannot give insides on the meteorological parameters changes due to urbanization. The sensible heat flux is function of the incoming radiation and specific properties of the surface. In this sense, if we consider that the incoming radiation is not significantly different at places located on 100 km distance, the sensible heat flux can be use as some indicator of urbanization, but cannot be measure of extension of Metro Manila influence. Temperature and precipitation are strongly affected from the sea breezes from Manila Bay (West of Manila) and Laguna Bay (South of Manila) at Metro Manila. All considering rural areas are located inland and represent totally different local conditions. So these two indicators are not appropriate to describe the effect on urbanization, as the effects of sea breeze circulation and very likely created convergent zone above the Metro Manila during the day, cannot be excluded and separated from the effect of urban area existence. More appropriate indicators can be the latent and ground heat fluxes produced by the model.  

Also largest urban heat islands (UHI) are likely to occur on calm clear nights following cloudless days with weak winds. This combination produces large amplitude waves of surface and air temperature in both urban and rural areas. Air masses and fronts, that are associated with synoptic or mesoscale systems, like land- and sea breezes, mountain-and-valley winds or Northeast Monsoon in this case, can completely overwhelm and destroy an UHI. They can also create spurious ‘heat-’or ‘cool islands’ when the heat island magnitude is derived from urban-rural pairs that are differentially affected at times when the advective effect is present at one station but not the other (see T. R. Oke, G. Mills, A. Christen, J. A. Voogt, “Urban climates”, Cambridge University Press, Sep 14, 2017). My suggestion is to extract periods with appropriate conditions for UHI investigation, or at least exclude the wet monsoon season.

Specific comments:

1.       Line 35: More recent document is available (2018 Revision of World Urbanization Prospects, UN DESA).

2.       I miss description of the modelling domains. How many domains are used in the nesting procedure? Add a sentence.

3.       Verification of the model results is necessary. Making quantitate conclusions based only on modelling output is inappropriate. The reader should be convinced first that the model simulate correctly real meteorological conditions.

4.       U.S. Geological Survey (USGS) Global Land Cover Characteristics data are based on 1-km Advanced Very High Resolution Radiometer (AVHRR) data spanning April 1992 through March 1993. This old database very often do not represent correctly fast developed urban areas during the last decades. Check if USGS urban and build-up category covers correctly the Metro Manila area.

5.       Lines 204-2009: The statements contradict each other and with lines 74-76 (The study period was chosen because major infrastructures were constructed within Metro Manila in this period. More infrastructures are currently being built so 75 current study is very significant.)

6.       Do authors expect that the urbanization of Metro Manila will affect the upward heat flux, temperature and rainfall at 100 km?

7.       Commonly used units for the heat flux is watts per square metre (Wm−2) and WRF model output is in the same units. Why the authors converted to J/m2?

Author Response

Please find the response attached.

Round 2

Reviewer 1 Report

The author(s) addressed the most of the comments successfully. 

Author Response

Again, we would like to special mention the reviewers for valuable comments and suggestions for the improvement of our article.

Reviewer 2 Report

The authors made some corrections mainly related to the specific comments, but unfortunately I cannot see substantial improvement of the manuscript. The main focus based on the presented results should be on climate not on urban heat island (UHI) and urbanization. If they change the title and rewrite the manuscript with focus on climate, the paper can be suitable for publication. There is a problem with the methodology they used and the experiments that have been carried out that is not applicable to study related to UHI. My major concerns are:

1.       Output on every three hours is not sufficient to study the processes related to UHI, as there is drastic changes in the properties especially in transition periods around sunset and sunrise, which are not captured with the model outputs.

2.       Simulations with 9 km grid spacing is inappropriate for this type of study, as very detail description is necessary to investigate effects of land use. The authors must go to 1km resolution (two more nested domains). Lines 67-68: “Due to its limited size, coastlines of Manila Bay, Laguna Bay, and Pasig River, and green parks were converted to high-rise building (commercial/residential), asphalted roads.” – This is result from the used coarse resolution, and artificially adding urban properties to the land cover increases a possible effect and compromise the findings in this study.

3.       Temperature and precipitation are strongly affected from the sea breezes from Manila Bay (West of Manila) and Laguna Bay (South of Manila) at Metro Manila. Investigating UHI you need comparison between areas with similar conditions. All considering rural areas are located inland and represent totally different local conditions. In this sense the study of diurnal temperature range and conclusions made are not appropriate.

4.       In the quoted paper “Impacts of urban expansion and future green plantingon summer precipitation in the Beijing metropolitan area” the conclusion is “…summer and annual precipitation in the Beijing area has decreased with urban expansion…” In this paper the rainfall is highest compared with all the sites having similar climate classification - Lines 255-257:“The 11-year average rainfall of Metro Manila is 172.46 mm for this season and is the highest compared with all the sites having similar climate classification. This is corresponding to 8% to 38%, 15% to 41%, 51% to 64%and 33% to 41% higher than the rainfall values in the groups of 25km, 50km, 75km and 100km, respectively.” The difference in rainfall values, registered at different locations, is more likely result from the distance to the ocean and local topography, than due to urban effects. The urban area affect the rainfall mainly through generated aerosols due to human activities that play role of condensation nuclei, so called secondary effect, which is not considered in this study.

5.       I am not satisfy with the answer “Upward heat flux was generated in the study.” The variable HFX from the WRF output (described as upward heat flux in the User’s Guide) is a sensible heat flux between the surface and the first model level, and the flux is a constant in the surface sub-layer of the Planetary Boundary Layer according the Monin-Obukhov theory, applied in WRF model also. The Surface Energy Balance equation is the balance of net radiation and the sum of ground heat flux and the two turbulent heat fluxes that exchange energy between the surface and atmosphere – the sensible heat flux and the latent heat flux (Oke T R, Mills G, Christen A, Voogt J (2017) Urban climates. Cambridge University Press, UK). The same Surface Energy Balance equation is used in WRF model. A surface layer parameterization provide the surface (bulk) exchange coefficients for momentum, heat, and water vapor used to determine the flux of these quantities between the land surface and the atmosphere (A Description of the Advanced Research WRF Version 3, NCAR/TN–475+STR, NCAR TECHNICAL NOTE, 2008). All described above show that the authors use a sensible heat flux, which can be indicator of urbanization, but cannot be measure of extension of Metro Manila influence and cannot affect the surrounding areas. It is enough to compare with one location in rural area with similar conditions. My recommendation was using the latent heat flux, instead of precipitation, as this flux can describe better effects of urbanization by substantial decrease in city areas.

Author Response

Again, we would like to special mention the reviewers for valuable comments and suggestions for the improvement of our article.

Listed in the tables are the revisions that we have made. Please see attached.

Round 3

Reviewer 2 Report

I do not have more suggestions.